# Toxins from Animal Venoms as a Potential Source of Antimalarials: A Comprehensive Review

**DOI:** 10.3390/toxins15060375

**Published:** 2023-06-03

**Authors:** Zeca M. Salimo, André L. Barros, Asenate A. X. Adrião, Aline M. Rodrigues, Marco A. Sartim, Isadora S. de Oliveira, Manuela B. Pucca, Djane C. Baia-da-Silva, Wuelton M. Monteiro, Gisely C. de Melo, Hector H. F. Koolen

**Affiliations:** 1Programa de Pós-Graduação em Medicina Tropical, Universidade do Estado do Amazonas, Manaus 69040-000, Brazil; zecasalimoo@gmail.com (Z.M.S.); alineadriaoam@gmail.com (A.A.X.A.); marcosartim@hotmail.com (M.A.S.); manupucca@hotmail.com (M.B.P.); djane.claryss@gmail.com (D.C.B.-d.-S.); wueltonmm@gmail.com (W.M.M.); 2Fundação de Medicina Tropical Dr. Heitor Vieira Dourado, Manaus 69040-000, Brazil; 3Grupo de Pesquisa em Metabolômica e Espectrometria de Massas, Universidade do Estado do Amazonas, Manaus 69065-001, Brazil; amr.bio18@uea.edu.br; 4Setor de Medicina Veterinária, Universidade Nilton Lins, Manaus 69058-030, Brazil; andrelima1701@gmail.com; 5Programa de Pós-Graduação em Biodiversidade e Biotecnologia—Rede BIONORTE, Universidade do Estado do Amazonas, Manaus 69065-001, Brazil; 6Pro-Reitoria de Pesquisa e Pós-Graduação, Universidade Nilton Lins, Manaus 69058-030, Brazil; 7Departamento de Ciências BioMoleculares, Faculdade de Ciências Farmacêuticas de Ribeirão Preto, Universidade de São Paulo, Ribeirão Preto 14040-903, Brazil; isadora_so@yahoo.com; 8Faculdade de Medicina, Universidade Federal de Roraima, Boa Vista 69317-810, Brazil; 9Programa de Pós-Graduação em Ciências da Saúde, Universidade Federal de Roraima, Boa Vista 69317-810, Brazil; 10Faculdade de Farmácia, Universidade Nilton Lins, Manaus 69058-030, Brazil; 11Instituto Leônidas e Maria Deane, Fundação Oswaldo Cruz, Manaus 69057-070, Brazil; 12Programa de Pós Graduação em Ciências Farmacêuticas, Universidade Federal do Amazonas, Manaus 69080-900, Brazil

**Keywords:** malaria, *Plasmodium*, antimalarials, resistance to antimalarials, animal venom toxins

## Abstract

Malaria is an infectious disease caused by *Plasmodium* spp. and it is mainly transmitted to humans by female mosquitoes of the genus *Anopheles*. Malaria is an important global public health problem due to its high rates of morbidity and mortality. At present, drug therapies and vector control with insecticides are respectively the most commonly used methods for the treatment and control of malaria. However, several studies have shown the resistance of *Plasmodium* to drugs that are recommended for the treatment of malaria. In view of this, it is necessary to carry out studies to discover new antimalarial molecules as lead compounds for the development of new medicines. In this sense, in the last few decades, animal venoms have attracted attention as a potential source for new antimalarial molecules. Therefore, the aim of this review was to summarize animal venom toxins with antimalarial activity found in the literature. From this research, 50 isolated substances, 4 venom fractions and 7 venom extracts from animals such as anurans, spiders, scorpions, snakes, and bees were identified. These toxins act as inhibitors at different key points in the biological cycle of *Plasmodium* and may be important in the context of the resistance of *Plasmodium* to currently available antimalarial drugs.

## 1. Introduction

Malaria remains a major public health problem [1]. In 2021, 247 million cases of malaria were registered worldwide, an increase from 245 million in 2020, with an estimated 619,000 deaths, the most vulnerable groups being children under 5 years of age, pregnant women and patients with HIV/AIDS. This increase was particularly evident in Africa (95%) [1]. Malaria is a potentially dangerous acute febrile infectious disease caused by *Plasmodium* spp. (species), which is transmitted to humans through the bite of an infected female *Anopheles* mosquito [2]. Currently, seven species are known to cause malaria in humans in different areas of the world (Table 1): *Plasmodium falciparum*, *Plasmodium vivax*, *Plasmodium knowlesi*, *Plasmodium ovale*, *Plasmodium malariae*, *Plasmodium cynomolgi* and *Plasmodium simium.*

*P. falciparum* malaria cases are most prevalent in the African region (mainly sub-Saharan Africa), Southeast Asia, the Eastern Mediterranean, and the Western Pacific region [1]. *P. vivax* is the dominant malaria species in much of Asia-Pacific, the Horn of Africa, and Central and South America, and caused 4.5 million cases worldwide in 2020 [3]. *P. ovale* is usually described as being limited to tropical Africa, the Middle East, Papua New Guinea, and Irian Jaya in Indonesia [4]. *P. knowlesi* has been reported in Malaysian Borneo; though cases have also been reported in Thailand, Myanmar, China, the Philippines, and Singapore [1]. *P. malariae* has been reported in Africa, the Southeastern Pacific and South America, *P. simium* in South America, and *Plasmodium cynomolgi* in Peninsular Malaysia, the Northern Sabah Kapit district in Sarawak and Malaysian Borneo (Table 1).

The containment of cases and progression towards the elimination of human malaria is related to adequate and timely treatment, in addition to vector control, mainly using insecticides [1,5]. Different antimalarials are currently available (Table 1 and Table 2) and are effective to some extent [1]. However, resistance to available antimalarials has been increasingly reported, and has become an important barrier to malaria elimination [1,6,7,8]. The discovery of new molecules or toxins may lead to medicines with greater antimalarial activity. As such, this review identifies and describes animal venom toxins with potential antimalarial activity.

## 2. *Plasmodium* Species Causing Human Malaria

The etiological agents of malaria are the protozoans of the genus *Plasmodium*, which are transmitted to the vertebrate host through the bite of infected *Anopheles* female mosquitoes [9]. More than 120 species of *Plasmodium* are known, but only 7 species of them are described as being capable of infecting humans, and *P. malariae*, *P. vivax*, and *P. falciparum* are the most common species [10,11]. *P. falciparum* is responsible for most of the severe cases and about 99% of malaria-associated deaths worldwide [1]. *P. vixax* is a predominant species in the Americas, causing 75% of malaria cases [1]; it can also cause severe cases, similarly to *P. falciparum* [12]. *P. knowlesi*, *P. simium* and *P. cynomolgi* are transmitted from primates to humans; however, the prevalence and clinical impact of these species are unclear, although the first species can cause severe manifestations [13,14]. *P. malariae* and *P. ovale* cause uncomplicated malaria, although they may sometimes be associated with other complications [13]. 

Female mosquitoes of the genus *Anopheles* are the vectors of *Plasmodium* spp. [1]. *Anopheles* are insects of great epidemiological importance [15]. Approximately 3500 mosquito species, grouped into 41 genera, are known. The genus *Anopheles* has a wide geographic distribution, and Antarctica is the only place that it is not found. This genus consists of about 430 species and, of these, only about 70 species are natural transmitters of malaria [15,16,17,18,19,20,21]. The *Plasmodium*-vector interaction is complex and there are factors (invasion of the intestinal cells of mosquitoes, ookinete escape, ookinete development time, vector immune response, among others, for example) that determine the specificity in the ecological relationship between vector-plasmodium and the geographical distribution of cases [22,23,24] (Table 1).

**Table 1 toxins-15-00375-t001:** Distribution of *Plasmodium* species on the continents of the globe, their respective vectors and recommended treatments.

Parasite	Vector	Location, Continent	Treatment	Ref.
*P. vivax*	*Anopheles albimanus* *Anopheles albitarsis* *Anopheles aquasalis* *Anopheles darlingi* *Anopheles freeborni* *Anopheles marajoara* *Anopheles nuneztovaris* *Anopheles pseudopunctipennis* *Anopheles quadrimaculatus* *Anopheles cruzzi* *Anopheles bellator* *Anopheles brasiliensis* *Anopheles calderoni* *Anopheles triannulatus* *Anopheles neivai* *Anopheles deaneorum* *Anopheles oswaldoi* *Anopheles argyritarsis* *Anopheles dunhami*	Central and South America	CQ + PQ AS + PQ CQ + TQ	[1,15,25,26,27,28,29]
	*Anopheles annularis* *Anopheles aconitus* *Anopheles subpictus*	*South and Southeast Asia and Asia-Pacific*	CQ AL DHA + PPQ AS + PY	[1,15,28]
	*Anopheles stephensi*	Africa	CQ DHA + PPQ	[1,15,26,27]
*P. falciparum*	*Anopheles arabiensis* *Anopheles funestus* *Anopheles gambae* *Anopheles stephensi* *Anopheles melas* *Anopheles merus* *Anopheles moucheti* *Anopheles nili*	Africa	AL AS + AQ AS + PY DHA + PPQ	[1,15,20,30]
	*Anopheles farauti* *Anopheles Kiliensis* *Anopheles punctulatus* *Anopheles dirus* *Anopheles minimus* *Anopheles lesteri* *Anopheles sinensis* *Anopheles balabacensis* *Anopheles barbirostris*	Asia	AL AS + MQ AS + PY AS + SP AS + SP DHA + PPQ + PQ	[1,15,30]
	*Anopheles atroparvus* *Anopheles labranchiae* *Anopheles messeae* *Anopheles sacharovi* *Anopheles sergentii* *Anopheles superpictus*	Mediterranean	AL AS + SP DHA + PPQ	[1,15,31]
	*Anopheles flavirostris* *Anopheles koliensis* *Anopheles lesteri* *Anopheles leucosphyrus* *Anopheles maculatus* *Anopheles punctulatus* *Anopheles sinensis* *Anopheles sundaicus*	Western Pacific	AL AS + PY DHA + PPQ AS + MQ	[1,15,30]
	*Anopheles darlingi* *Anopheles deaneorum*	Central and South America	AL AS + MQ	
*P. malariae*	*Anopheles stephensi*, *Anopheles gambiae*	Africa	CQ + PQ	[1,15,29,32,33,34]
	*Anopheles freeborni* *Anopheles dirus*	Southeastern Pacific		
	*Anopheles darlingi*	South America		
*P. ovale*	*Anopheles gambiae* *Anopheles funestus*	Africa	CQ + PQ and/or ART	[1,15,34]
	*Anopheles flavirostris* *Anopheles koliensis* *Anopheles lesteri*	Western Pacific		
*P. simium*	*Anopheles nyssorhinchus* *Anopheles Kerteszia*	South America	CQ + PQ	[1,11,15]
*P. knowlesi*	*Anopheles hackeri* *Anopheles latens* *Anopheles sundaicus* *Anopheles dirus* *Anopheles hacker* *Anopheles cracens* *Anopheles introlatus*	Malaysia	CQ + PQ	[1,15]
*P. cynomolgi*	*Anopheles balabacensis*	Malaysia	CQ + PQ	[1,14,35]

Abbreviations: CQ—chloroquine; PQ—primaquine; AS—artesunate; ART—artemisinin; MQ—mefloquine; AL—artemether-lumefantrine; AQ—amodiaquine; DHA—dihydroartemisinin; PPQ—piperaquine; PY—pyronaridine; SP—sulfadoxinepyrimethamine.

## 3. *Plasmodium* Life Cycle and Pathogenicity

The life cycle of *Plasmodium* is heteroxenous and occurs in the mosquito vector and in vertebrates (e.g., humans) (Figure 1). The infection begins with the bite of a mosquito infected with *Plasmodium* sporozoites (Figure 1A) [36]. The sporozoites are inoculated during the blood meal (Figure 1A). They slip through the dermis and subsequently enter the blood circulation and migrate to the hepatic sinusoids to invade the hepatocytes [37] (Figure 1B). After the invasion, the sporozoites divide asexually by pre-erythrocytic schizogony and form pre-erythrocytic trophozoites that multiply, giving rise to tissue schizonts (Figure 1B) [25,37]. The duration of pre-erythrocytic schizogony varies according to the infectious species (8 to 27 days for *P. vivax*, 8 to 25 days for *P. falciparum*, 9 to 17 days for *P. ovale*, 15 to 30 days for *P. malariae*, 9 to 12 days for *P. knowlesi* and still unknown for *P. simium* and *P. cynomolgi*) [38]. At this stage, *P. vivax* and *P. ovale* form hypnozoites, which are latent forms of the parasite that are responsible for relapses of the disease months or years later [36,37]. 

At the end of the first phase (Figure 1B), also called the exoerythrocytic or tissue stage, each infected hepatocyte releases thousands of exoerythrocytic merozoites. The number varies according to the species (about 2000 merozoites when the infection is by *P. malariae*; 10,000 when due to *P. vivax*; 40,000 when due to *P. falciparum* and 15,000 when due to *P. ovale*, though the quantity is still unknown for *P. simium* and *P. cynomolgi*) [39]. Merozoites released from hepatocytes invade red blood cells (Figure 1C), which initiates the erythrocytic phase. *P. vivax* preferentially invades young erythrocytes, *P. falciparum* invades erythrocytes in any evolutionary phase, while *P. malariae* invades old erythrocytes [39]. After invading the erythrocytes, merozoites divide asexually giving rise to ring forms, trophozoites, and young and mature schizonts [36]. During a period that varies from 48 to 72 h, the parasite develops inside the erythrocytes until it causes their rupture, thus releasing new merozoites that will invade new erythrocytes [39] (Figure 1C). The rupture and consequent release of merozoites into the bloodstream is clinically translated by the onset of the malarial paroxysm, which will be repeated at the end of the new cycle [39]. This cycle of invasion-multiplication-release-invasion is repeated [36]. After a period of asexual replication, some merozoites differentiate into male and female gametocytes (Figure 1D), which mature without cell division and become infectious to mosquitoes [39] (Figure 1E). 

Mosquitoes become infected with *Plasmodium* during a blood meal from an infected host (Figure 1E). In the vector, the sexual reproduction (sporogony) of the malaria parasite occurs in the mosquito’s stomach after the differentiation of gametocytes into gametes and their fusion, with the formation of the zygote [39] (Figure 1E). The zygote transforms into a mobile form (ookinete) that transposes the peritrophic matrix and then migrates to the midgut wall of the insect and forms the oocyst, within which the sporozoites will develop [36,39] (Figure 1E). The sporozoites produced in the oocysts are released into the insect’s hemolymph and migrate to the salivary glands, from where they are transferred to the blood of the human host during the blood meal [39] (Figure 1A). The time required for completion of the sporogony cycle in insects varies depending on the *Plasmodium* species and the temperature, though it generally takes around 10 to 20 days [36].

During the life cycle of *Plasmodium*, it can invade the red blood cells to feed on hemoglobin. As it feeds, it ruptures the cells, releasing red blood cells and parasite debris, including malarial pigment (hemozoin) and glycophosphatidylinositol, called malarial toxin, thus causing the symptoms [40]. The more erythrocytes that are infected and rupture and release putative malarial toxins, the greater the pathogenesis or severity of malaria [41,42]. Putative malarial toxins activate peripheral blood mononuclear cells and stimulate the release of cytokines with a consequent systemic inflammatory response [41]. The balance between pro-inflammatory and anti-inflammatory cytokines, chemokines, growth factors, and effector systems determines the severity of the disease [41]. The pathogenicity of malaria also depends on the individual’s immunological characteristics, the genetic aspects of the parasite and host, previous exposure to infection, age, and nutritional, geographic and socioeconomic factors [43]. Clinical complications of malaria include severe anemia, acute renal failure, acute pulmonary edema, algid malaria, and cerebral malaria, and they can be avoided through early diagnosis and treatment [1,42]. 

## 4. Malaria Treatment

Antimalarial drugs can act by interrupting the multiplication of the parasite and, consequently, the inhibition of malarial infection by affecting different stages of the parasite throughout the cycle [44]. Antimalarial drugs target (a) the parasite asexual erythrocytic stages, (b) tissue schizonticides by targeting hypnozoites and gametocytocides, which destroys the sexual forms of the parasite in the bloodstream, thus preventing the transmission of malaria to the mosquito, and (c) the sporontocides, which prevents or inhibits the formation of oocysts and malaria sporozoites in the infected mosquito [45]. For the adequate treatment of malaria, the following is necessary: identification of the infectious species; identification of the susceptibility of the *Plasmodium* species to the drug; and the clinical status of the patient [46]. The antimalarials in current use are summarized in Table 1 and Table 2.

The available antimalarials are categorized into seven classes: (a) sesquiterpene lactone endoperoxides compounds: artemisinin (ART), dihydroartemisinin (DHA), artesunate (AS) and artemether (AR); (b) 4-aminoquinolines: chloroquine (CQ), amodiaquine (AQ), pyronaridine (PY), piperaquine (PPQ) and naphthoquine (NQ); (c) arylaminoalcohols: quinine (QUIN), mefloquine (MQ), halofantrine (HF) and lumefantrine (LR); (d) 8-aminoquinolines: primaquine (PQ) and tafenoquine (TQ); (e) antifolates: proguanil (PG), pyrimethamine (PMT) and sulfadoxine (SULF); f) naphthoquinones: atovaquone (ATQ) and (g) antibiotics: clindamycin (CLI), doxycycline (DOX) and tetracycline (TC) (Table 2, Figure 2). Their mechanisms of action are summarized in Table 2.

However, in some regions, the first line of treatment is via artemisinin combination therapies (ACTs) + PQ/TQ due to CQ resistance [47,48,49,50]. Treatment for *P. falciparum* infections is performed by the combination of AR and LR or with MQ or QUIN, DOX and PQ [25]. For mixed infections caused by *P. falciparum* and *P. vivax* (or *P. ovale*), treatment should include a blood schizonticidal drug that is effective for *P. falciparum*, associated with PQ (tissue schizonticidal). If the mixed infection is *P. falciparum* and *P. malariae*, treatment should be directed towards *P. falciparum* only [25]. With regard to the treatment of severe and complicated malaria, the aim is to prevent the patient from dying, so doctors must follow a rigid treatment scheme that consists of modulating the dosage, and these schemes are already defined in the strategic plan for the treatment of malaria in each country. However, the WHO recommends the administration of injectable artesunate (intramuscular or intravenous), followed by an ACT-based treatment as soon as the patient can take oral medications. If injectable treatment is not possible, the patient should immediately be given artesunate intrarectally and transferred as soon as possible to a suitable site for full parenteral treatment [1,25] (Table 2).

**Table 2 toxins-15-00375-t002:** Available drugs, their chemistry classes, origin, target species, points where it is active and mechanism of action.

Class	Name	Origin of the Drug	*Plasmodium* spp.	Active against Stages	Mechanism of Action	Ref.
Sesquiterpene lactone endoperoxides	ART	*Artemisia annua* L.	*P. falciparum* and *P. vivax*	All	Protein metabolism	[50,51,52,53]
	AS	Semi-synthetic derivative of artemisinin	*P. vivax*	All		[50,54,55,56]
	AR		*P. falciparum and P. vivax*	All		[50,51,57]
	DHA		*P. falciparum and P. vivax*	All	Not very well known. Probably protein metabolism	[46,51]
4-aminoquinolines	CQ	Synthetic analogue of quinine	*P. vivax*	Blood stages (trophozoite and schizont)	Digestion of hemoglobin	[50,54,55,56,57,58,59]
	AQ		*P. vivax*			[50,55,60,61]
	PPQ		*P. falciparum* and *P. vivax*		Not very well known. Probably digestion of hemoglobin	[46,56,59,60]
	PY		*P. falciparum* and *P. vivax*			[46,51,56,62]
	NQ		*P. falciparum* and *P. vivax*			[46,63]
Arylaminoalcohols	QUIN	*Cinchona calisaya* L.	*P. falciparum* and *P. vivax*	Blood stages (trophozoite and gametocytes)	Digestion of hemoglobin	[50,64]
	MQ	Synthetic derivative of quinoline	*P. falciparum* and *P. vivax*	Blood stages (trophozoite, schizont and gametocytes)		[50,52,56]
	LR	Synthetic derivative of fluorene	*P. falciparum* and *P. vivax*		Not very well known. Probably digestion of hemoglobin	[50,64]
	HF	Synthetic derivative of fluorene	*P. falciparum*		Digestion of hemoglobin	[65,66]
8-Aminoquinolines	PQ	Synthetic, 8-aminoquinoline derivative	*P. vivax*		Forms quinoline-quinone metabolites that act as oxidants	[46,67]
	TQ	Synthetic analogue of primaquine	*P. vivax and P. falciparum*		Interferes with the polymerization of the heme group	[46,60]
Antifolates	PMT	Synthetic derivative of ethyl-pyrimidine	*P. falciparum* and *P. vivax*	Blood, liver (schizont) and mosquito (oocysts) stage	Inhibits dihydrofolate reductase enzyme and blocks plasmodium DNA synthesis	[46,68]
	SULF		*P. vivax*	Blood and liver (schizont) stage	Inhibits parasite dihydropteroate synthetase	[46,69]
	PG	Biguanide	*P. falciparum* and *P. vivax*	Blood (gametocyte) and liver (shizont) stages	Inhibits parasite dihydrofolate reductase	[46,50]
Naphthoquinones	ATQ	Synthetic hydroxynaphthoquinone	*P. falciparum* and *P. vivax*		Inhibits electron transport in the mitochondria of parasites	[46,58]
Antibiotics	CLI	Semisynthetic of lincomycin	*P. falciparum*	All blood stages	Inhibits protein synthesis in the *Plasmodium* apicoplast	[46,70]
	DOX	Semi-synthetic of tetracycline	*P. falciparum*			[46,64,70]
	TC	Semi-synthetic of chlortetracycline	*P. falciparum*			[50,70]

Abbreviations: ART—artemisinin; DHA—dihydroartemisinin; AS—artesunate; AR—artemether; CQ—chloroquine; AQ—amodiaquine; PY—pyronaridine; PPQ—piperaquine; NQ—naphthoquine; QUIN—quinine; MQ—mefloquine; HF—halofantrine; LR—lumefantrine; PQ—prima-quine; TQ—tafenoquine; PG—proguanil; PMT—pyrimethamine; SULF—sulfadoxine; ATQ—atovaquone; CLI—clindamycin; DOX—doxycycline; TC—tetracycline.

### Resistance to Antimalarials

Drug resistance to antimalarials threatens the control and elimination of malaria [71]. *P. falciparum* has developed resistance to all currently used antimalarials, but there is a variation in geographic distribution and degree of resistance (Table 3). The most resistant parasites are found in Southeast Asia (Table 3). Resistance is lowest in *P. vivax*, although resistance to CQ is found throughout Indonesia and Papua New Guinea [13]. CQ-resistance has spread much more slowly in *P. vivax* populations when compared to *P. falciparum* [48]. Possible causes include a small parasite load in *P. vivax* infections, early gametocytogenesis and transmission before resistant clones are selected in the host under drug pressure, and the very high genetic diversity in natural populations of *P. vivax* [48]. 

The resistance of *Plasmodium* spp. to antimalarial drugs has been mainly associated with genetic mechanisms (Table 3), thus several studies on molecular markers have identified and tracked genes expressed by the parasite, as well as key mutations [1]. Resistance to CQ is associated with the development of a transporter for CQ encoded by the CQ resistance transporter orthologue gene of *P. vivax* (*pvcrt-o*) or CQ resistance transporter gene of *P. falciparum* (*pfcrt*), which prevents its absorption and metabolization in the parasite’s food vacuoles (Table 3). The *P. vivax* multidrug resistance gene 1 (*pvmdr1*) and *P. falciparum* multidrug resistance gene 1 *(pfmdr1*) were also associated with resistance to CQ in *P. vivax* and *P. falciparum*, respectively (Table 3). Resistance to MQ was identified by the amplification of *pvmdrl*/*pfmdr1* and over-expression of *p*-glycoprotein homologue 1 protein (*pgh1*) for *P. vivax* and *P. falciparum*, and the same strategy regarding HF and QUIN is used for *P. falciparum* (Table 3). In regard to resistance to AQ, *P. vivax* or *P. falciparum* reduce the affinity of binding of the competitive inhibitor to dihydrofolate reductase *(dhfr)*, and *P. falciparum* uses the same strategy regarding PG and PMT (Table 3). In terms of resistance to ART, AS, AR, and LR, although not very well known, it has been associated with phenotypes expressed in trophozoite ring stages during the *P. falciparum* cycle and mutation of the Kelch protein 13 *(pfk13)* in specific sequences of the domain-containing protein 1 (*btb-poz*) and six kelch domains that somehow impede parasite protein metabolism (Table 3).

**Table 3 toxins-15-00375-t003:** *Plasmodium* drug resistance, the location where documented and its resistance mechanism.

Parasite	Drug Resistance	Location Where *Plasmodium* Resistance Has Been Documented	Resistance Mechanism	Ref.
*P. vivax*	CQ	Asia and Oceanian (Papua New Guinea and Indonesia), South America (Brazil) Africa (Madasgascar and South and Southeast Asia (India, Myanmar, Nepal, and Thailand)	Develops a transporter for chloroquine, encoded *pvcrt-o* Mutation of *pvmdr1* and *pvdhfr*	[72,73,74,75,76]
	MQ	Southeast Asia (Western border of Thailand) and South America (Brazil)	Amplification of *pvmdrl* and over-expression of *pghl*	[75,77]
	AQ	Southeast Asia (Western border of Thailand)	Reduced *dhfr* affinity	[75,76]
*P. falciparum*	ART	Southeast Asia and East Asia (Thailand, Vietnam, Myanmar, Laos, China), and Sub-Saharan Africa	Not well known Mutation of *pfk13* in specific sequences of the BTB-POZ domains and six kelch domains (probably)	[78,79]
	AS	Southeast Asia (Western Cambodia)		[78]
	AR	Southeast Asia and Sub-Saharan Africa		[78,79]
	LR	Southeast Asia and Sub-Saharan Africa		[77,78]
	CQ	Southeast Asia (Western border of Thailand), Africa (Sub-Saharan Africa), South America (Brazil)	Develops a transporter for chloroquine, encoded *pfcrt;* Mutation of *pfmdr1* and *pfdhfr*	[75,80]
	AQ	Western border of Thailand	Reduced affinity for binding of the DHFR competitive inhibitor	[76]
	QUIN	Southeast Asia (Thailand, Thai Myanmar and Thai-Cambodian borders)	Amplification of *pfmdrl* and *pghl* overexpression	[76,81,82]
	PG	Southeast Asia (Thailand, Thai Myanmar and Thai-Cambodian borders)	Reduced affinity for binding of the *dhfr* competitive inhibitor	[76,81]
	MQ	Southeast Asia (Western border of Thailand, Thai-Myanmar and Thai-Cambodian borders) South America (Brazil)	Amplification of *pfmdrl* and *pghl* over-expression	[75,76,81,82,83,84]
	HF	Southeast Asia (Thailand)	Amplification of *pfmdrl* and *pghl* over-expression	[80,81]
	SULF	Southeast Asia (Western border of Thailand, Thai-Myanmar and Thai-Cambodian borders)	*dhps* mutations	[75,76,81]
	PMT	Southeast Asia (Western border of Thailand, Thai-Myanmar and Thai-Cambodian borders)	Reduced affinity for binding of the *dhfr* competitive inhibitor	[75,76,81]

Abbreviations: CQ—chloroquine; MQ—mefloquine; AQ—amodiaquine; ART—artemisinin; AS—artesunate; AR—artemether; LR—lumefantrine; QUIN—quinine; PG—proguanil; HF—halofantrine; SULF—sulfadoxine; PMT—pyrimethamine; *pvcrt-o*—*P. vivax* chloroquine resistance transporter orthologue gene; *pvdhfr*—*P. vivax* dihydropteroate reductase; *pvmde1*—*P. vivax* multidrug resistance gene 1; *pgh*1—*P*-glycoprotein homologue 1 protein; DHFR—dihydrofolate reductase; *pfk13*—*P. falciparum* Kelch protein 13; *pfcrt*—*P. falciparum* chloroquine resistance transporter gene; *pfmdr1*—*P. falciparum* multidrug resistance gene 1; *pfdhfr*—*P. falciparum* dihydropteroate reductase; *dhfr*—competitive inhibitor to dihydro-folate reductase; *dhps*—deoxyhypusine synthase; BTB-POZ—domain-containing protein 1.

## 5. Antiplasmodial Toxins

In view of the morbidity and mortality rates associated with malaria, the high global distribution of malaria, the resistance of *Plasmodium* to the drugs used in the treatment, and the resistance of the vector (*Anopheles* spp.) to conventional insecticides, there is a need to seek alternatives for the development of new drugs for combating and controlling the transmission of malaria [85]. 

Animal venoms are a complex mixture that contains many proteins, enzymes, peptides, and small molecules [86,87]. Depending on the taxonomic group, the toxins present in venoms have different modes of action and can be used for defense against predators and pathogens in the environment [88,89,90]. It is known that animal venoms have a range of molecules with antimicrobial properties, thus making them an important resource for the investigation of compounds with antimalarial potential [90].

Over the years, several studies have been carried out to demonstrate the antiplasmodial effect of toxins from different taxonomic groups (Table 4). Snake toxins are much more commonly studied for this purpose; however, there are studies for toxins from arachnids, scorpions and bees [91,92,93,94]. Although studies with toxins are mostly carried out with venom from animals, there are records of antiplasmodial activity for the secretion of some frog species, whose venom inoculation system is passive [95]. Among the toxins that are active against *Plasmodium*, these are mainly peptides and enzymes [96,97]. Crude extracts and fractions of venom extracts from some animals have also been tested and have demonstrated antiplasmodial activity [98].

Table 4 summarizes the toxins from venomous animals that have shown activity against *Plasmodium* spp. Isolated substances and total extracts and fractions of extracts that have shown antiplasmodial activity in studies are evidenced. Figure 3 shows some of the three-dimensional structures of molecules isolated from animal venoms and which have antiplasmodial activity.

### 5.1. Snake Toxins against Plasmodium

Snakes are animals that belong to the class Reptilia and the order Squamata, and there are an estimated 3970 species around the world [99]. During foraging, some snake species capture their prey and kill it by constriction, but others have venom-inoculating structures [100]. The toxins present in snake venoms have a complex of substances with neurotoxic, cardiotoxic and inflammatory capacity [101], and are efficient in killing certain types of prey. Searches for bioactive substances in snake toxins are extensively carried out for the formulation of new drugs, with captopril being known as a successful case. It is a drug used to control blood pressure whose active ingredient was isolated from the venom of the snake *Bothrops jararaca* [102]. It is known that several species present toxins that are mainly derived from proteins that have antibacterial, antifungal, antiviral and antiparasitic characteristics [90].

#### 5.1.1. Crude Snake Venoms

Extracts, fractions, enzymes, and some peptides found in snake toxins have antiplasmodial activity (Table 4). Terra et al. [103] showed that the crude venom extract of *Micrurus spixii* (Squamata, Elapidae) has great *P. falciparum* inhibitory power against its intraerythrocytic development (IC_50_ = 0.78 µg/mL). Hajialiani et al. [104] and Hajialiani et al. [105] tested *Naja naja oxiana* venom extracts against the parasites *P. falciparum* and *P. berghei*, respectively. In both studies, fraction 4, which was obtained from the crude extract of the venom of *Naja naja oxiana*, was used for anti-*Plasmodium* assays, and satisfactory inhibitory results were found (IC_50_ = 3.2 µg/mL) [104], as well as the interruption of parasitemia in 70%, 50% and 30% with the different concentrations of the fraction, 5, 2.5 and 1 mg/kg, respectively [105].

#### 5.1.2. Peptides from Snake Venoms

The inhibitory activities of some peptides isolated from snake venoms were tested against *Plasmodium* spp. [106,107]. Maluf et al. [106] isolated a cationic polypeptide with 42 amino acid residues (YKQCHKKGGHCFPKEKICLPPSSDFGKMDCRWRWKCCKKGSG), called crotamine, from *Crotalus durissus*, and tested its activity against *P. falciparum*. This substance inhibited the development of *P. falciparum*, presenting an IC_50_ of 1.87 µg/mL. Similarly, Fang et al. [107] tested the LZ1 peptide (VKRWKKWWRKWKKWV-NH2 CAS #) derived from the cathelicidin (polypeptide) of *Bungarus fasciatus* (Squamata, Elapidae) against *P. falciparum* and *P. berghei* in in vitro and in vivo assays. In the in vitro antiplasmodial assay, strong suppression was observed for *P. falciparum* (IC_50_ = 3.045 µM); and, in the in vivo assay, it was possible to observe expressive antimalarial activity, with 39% (4 mg/kg), 35% (8 mg/kg) and 24% (12 mg/kg). 

#### 5.1.3. Phospholipase A_2_ from Snake Toxins

Phospholipase A_2_ (PLA_2_) are a superfamily of enzymes known to have the ability to catalyze the hydrolysis of fatty acids at the *sn*-2 position to produce free fatty acids and lysophospholipids. They are small molecules of between 14 and 38 kDa that have between 5 and 8 disulfide bridges. This superfamily comprises a number of proteins, which are classified into 15 groups and 5 types: secreted, cytosolic, Ca^2+^ independent, acetyl hydrolases and lysosomal [108]. PLA_2_s have been detected in venoms from many snakes of the Elapidae and Viperidae families, and are mostly present in toxins from species of the genera *Bothrops* and *Crotalus* (Table 4).

Of the total number of studies that provided evidence of the inhibitory activity of snake venoms against *Plasmodium*, 68% corresponded to PLA_2_ (Table 4). The first study to test the antiplasmodial activity of a PLA_2_ derived from snake toxins was carried out by Zieler et al. [109]. The authors noted that PLA_2_ isolated from the venom of the eastern diamondback rattlesnake (*Crotalus adamanteus*) inhibited *Plasmodium gallinaceum* oocyst formation in *Aedes aegypti*. The reduction in the rate and intensity of infection was 29% and 24%, respectively. Despite being a notable result, in this study neither the average inhibitory concentration nor the minimum lethal concentration of the molecule was evaluated. Guillaume et al. [110] evaluated seven PLA_2_s from groups IA, IB, IIA and III against in vitro intraerythrocytic development of *P. falciparum*. Anti-*Plasmodium* activity was tested for toxins from the vipers *Agkistrodon halys* and *Vipera ammodytes*, which both have PLA_2_s belonging to group IIA, and from the elapids *Naja mossambica mossambica* and *Naja scutatus scutatus*, which have PLA_2_s from group IA. All PLA_2_s, from both groups, inhibited *Plasmodium* development (IC_50_ = 0.023 nM for *N. mossambica mossambica*, 2.6 nM for *N. scutatus scutatus*, 0.823 nM for *A. halys* and 2.8 nM for *V. ammodytes*). Castilho et al. [96] tested *Bothrops asper* venom against *P. falciparum* using the whole venom, a catalytically active PLA_2_ (fraction V) and a PLA_2_ homologue (fraction VI) due to its enzymatic activity. Fraction V had an IC_50_ of 1.42 µg/mL while fraction VI had an IC_50_ of 22.89 µg/mL and the whole venom had an IC_50_ of 0.13 µg/mL, thus demonstrating high inhibitory power. Quintana et al. [111] tested two fractions with PLA_2_s from the crotoxin complex and PLA_2_ crotoxin B. Fractions 1 and 2 containing PLA_2_s from the crotoxin complex, as well as crotoxin B, inhibited the intraerythrocytic development of *P. falciparum* (IC_50_ = 0.17, 0.76 and 0.6 µg/mL, respectively). PLA_2_ (BmajPLA2-II) isolated from *Bothrops marajoensis* venom showed inhibition of *P. falciparum* development (IC_50_ = 6.41 μg/mL) [112]. PLA_2_s isolated from the venom of another species of the *Bothrops* genus were also evaluated against *P. falciparum*. PLA_2_s (BdTX-I and BdTX-II) and the BdTX-III analogue isolated from *B. diporus* also showed inhibitory characteristics (IC_50_ = 2.44, 0.0153 and 0.5913 µg/mL, respectively) [113].

Simões-Silva et al. [114] analyzed the venom of *B. asper* and isolated and characterized five new PLA_2_ isoforms, four of them acidic and one with the basic form, which were grouped into two groups: Asp49-PLA and Lys49-PLA_2_-like, respectively. Two PLA_2_s (BaspAc-II and BaspAc-IV) showed activity against *P. falciparum* (IC_50_ = 2.46 and 0.019 µg/mL, respectively). Furthermore, the mixture of the two PLA_2_s that were active against *Plasmodium* (BaspAc-II and BaspAc-IV) showed activity that was ten times greater than when tested individually, with a fractional inhibitory concentration (FIC) of 0.498 µg/mL, thus demonstrating a synergistic effect. Most of the tests used to evaluate the effect of PLA_2_s used CQ-resistant strains of *Plasmodium*. The positive and significant results regarding *Plasmodium* inhibition show that the molecule may be an important alternative for disease control [114].

### 5.2. Toxins from Anurans against Plasmodium

The class Amphibia is divided into 3 orders, namely Gymnophiona (caecilians), Caudata (salamanders and newts) and Anura (toads, frogs, and tree frogs), the latter being the most representative of the class with approximately 7500 described species [115,116]. Anurans have an integument that is devoid of hair, feathers, or scales for protection, and have developed strategies to avoid water loss and infections by pathogens in the environment [115]. They are highly predated by both vertebrates and invertebrates at all stages of development and, therefore, have several characteristics that reduce the risk of predation [117]. Anurans have a vast framework of substances in their skin and, due to their integument being unprotected, these substances act not only to control infections by pathogens, but are also used to protect against predators [88,118]. Several substances have been tested against microorganisms and have shown antimicrobial characteristics [88]; however, studies that investigate antiplasmodial activity in anuran venoms are still scarce.

#### 5.2.1. Crude Anuran Venom

Antimicrobial assays involving crude extracts of anuran venoms are commonly performed [119]. Assays to assess antiplasmodial activities using crude extracts of anuran venoms have already been performed for bufonids [98,120]. The crude extract of *Rhinella marina* venom is active against *P. falciparum* (IC_50_ = 2.43 µg/mL) [120] and the venom of a bufonid (unspecified) showed good control of parasitemia against *P. berghei* [98]. The crude extracts tested showed inhibition of the development of trophozoites of both *Plasmodium* species [98,120].

#### 5.2.2. Steroids from Anuran Venoms

The bufadienolide known as telocinobufagin is present in the secretion of the bufonids *Rhinella marina* and *Rhaebo guttatus* [95]. Bufadienolides are steroids that are commonly found in the secretion of bufonids, and are known to be associated with intoxication processes in domestic animals, as well as being involved in chemical defense events [121]. The steroid telocinobufagin, isolated from the secretions of both toad species, showed antiplasmodial activity in vitro for *P. falciparum* by interfering with the development of trophozoites (IC_50_ = 1280 µg/mL) [95]. Three other bufadienolides, marinobufotoxin, marinobufagin, and bufalin, were isolated from the secretion of *R. marina* and showed antiplasmodial activity in vitro for *P. falciparum* by interrupting the development of trophozoites (IC_50_ = 5.31, 3.89 and 3.44 µg/mL, respectively) [122].

#### 5.2.3. Peptides from Anuran Venoms

Peptides are commonly a major component in anuran venoms and may be involved in communication [123] and defense events [124]. A large number of peptides present in the integument of anurans have an antimicrobial character and, for this reason, a series of studies have been carried out to test their activity against various microorganisms [125,126,127]. So far, only one peptide (phylloseptin-1), which was isolated from the tree frog *Phyllomedusa azurea* (Hylidae, Phyllomedusinae), has been tested for its activity against *P. falciparum* [128] and it showed the ability to inhibit the growth of trophozoites in in vivo experiments (MIC = 128 g/mL). Phylloseptin-1 is a peptide with 19 amino acid residues and presents an amidated C-terminal region (FLSLIPHAINAVSAIAKHN-NH_2_) [128]. Due to the high number of peptides already known and isolated from anuran venoms, it is suggested that more studies should be carried out to test their antiparasitic activity.

#### 5.2.4. Alkaloids from Anuran Venoms

Anurans, for the most part, are not capable of synthesizing alkaloids in their bodies; therefore, this metabolite is obtained from the ingestion of ants, termites and mites [124] Like peptides, alkaloids are involved in chemical defense processes and, in tests, they have demonstrated antiviral [129], antibacterial and antifungal activity [126]. Five families of anurans have alkaloids in their secretions. Dendrobatidae presents the largest number of individuals known to have alkaloids in their toxins [86,118], the other families being Mantellidae, Eleutherodactylidae, Bufonidae, and Myobatrachidae. The alkaloid dehydrobufotenine, isolated from the secretion of the bufonid *R. marina*, was tested for antiplasmodial activity and showed great inhibitory activity against the development of *P. falciparum* trophozoites (IC_50_ = 19.11 µg/mL) [122]. 

### 5.3. Spider Venom Toxins against Plasmodium Species

Spiders belong to the class Arachnida, order Araneae, and there are approximately 50,000 species around the world [130]. Venom glands are present in most spiders, but they are absent in the family Uloboridae [131]. The glands are located either in the chelicerae or under the carapace [131]; however, the toxic potential of venoms varies according to the species, since it is used not only as a defense mechanism, but also in hunting events [132]. They are venomous animals that can cause harm to humans, which is why some species are considered a public health problem [133]. Their toxins have a range of substances that present bioactive properties, some of which are used in the pharmaceutical industry to produce drugs and even serums [134]. Some species have had their venoms tested and have demonstrated a broad spectrum of antimicrobial activities [135]; however, studies that evaluate their antiplasmodial potential are still scarce. The peptides psalmopeotoxin I (PcFK1) and psalmopeotoxin II (PcFK2) were isolated from the venom of the tarantula *Psalmopeus cambridgei* (Araneae, Theraphosidae) and were tested against *Plasmodium* sp. PcFK1 has 33 amino acid residues in its primary sequence (ACGILHDNCVYVPAQNPCCRGLQRYGKCLVQV), while PcFK2 has 28 amino acid residues (RCLPAGKTCVRGPMRVPCCGSCSQNKCT). Both peptides were tested in vitro against *P. falciparum*. It was observed that both PcFK1 and PcFK2 showed antiplasmodial activity by inhibiting the development of *P. falciparum* trophozoites (IC_50_ = 1.59 and 1.15 µg/mL, respectively) [136].

### 5.4. Scorpion Venom Toxins against Plasmodium Species

Like spiders, scorpions are arachnids and belong to the class Arachnida. They are animals that have an elongated body and a venom inoculating device (telson) at the tip of the tail [137]. Currently, approximately 2200 species are known [94,138]. Toxins from some scorpion species have been tested against *P. falciparum*, *P. berghei* and *P. gallinaceum* (Table 4). Scorpine, isolated from the venom of *Pandinus imperator* (Scorpionidae), was evaluated against *P. berghei* [97]. Scorpine is a peptide with 75 amino acid residues (GWINEEKIQKKIDERMGNTVLGRMAKAIVHKMAKNEFQCMANMDMLGNCEKHCQTSGEKGYCHGTKCKCGTPLSY), and presents inhibitory activity against gametocytes and ookinetes of *P. berghei*, with a minimum inhibitory concentration (MIC) of 50 μM (gametocytes) and 30 μM (ookinetes). Meucin-24, isolated from *Mesobuthus eupeus* (Buthidae), is a peptide that has 22 amino acid residues (GRGREFMSNLKEKLVKEKMKNS) in its primary structure and presents antiplasmodial activity for *P. berghei* and *P. falciparum* [139]. Meucin-24 can inhibit the development (40% inhibition) of both parasites at concentrations between 10 and 20 μM, while meucin-25, on the other hand, showed inhibitory activity of 50% at the same concentrations. VmCT1-NH_2_ and its analogues [Arg]3-VmCT1-NH2, [Arg]7-VmCT1-NH_2_ and [Arg]11-VmCT1-NH_2_ isolated from *Vaejovis mexicanus* (Vaejovidae) venom were tested for activity against *P. gallinaceum*. VmCT1-NH_2_, [Arg]3-VmCT1-NH2 and [Arg]7-VmCT1-NH_2_ showed inhibitory capacity in the sporozoite phase (IC_50_ = 0.49, 0.57 and 0.51 µg/mL, respectively) [140]. 

### 5.5. Bee Venom Toxins against Plasmodium Species

Bees are insects that belong to the order Hymenoptera and the superfamily Apoidea. They are eusocial and present an organization at a hierarchical level of castes. They are highly appreciated not only for their ecological importance (they are excellent pollinators of plants), but also for production of a highly appreciated wax [141]. They have a stinger and are considered venomous [142]. In the study of antiplasmodial substances, bee venom toxins have been analyzed and have shown promise, with the main activities being attributed to substances that originate from proteins (Table 4).

#### 5.5.1. Peptides from Bee Venoms

Melittin and apamin were tested for their antiplasmodial activities. Melittin is a peptide that contains 26 amino acid residues in its primary structure (GIGAVLKVLTTGLPALISWIKRKRQQ) and is present in the venom of the bee *Apis mellifera* (Apidae) [94]. The first study to use synthetic melittin against malaria was carried out by Boman et al. [94] against *P. falciparum* and it was observed to inhibit the growth (in vitro) of the trophozoite and schizont forms at very low concentrations (MIC = 2 to 20 μM). Methyllin inhibited the intraerythrocytic growth of *P. falciparum* (IC_50_ = 10 µg/mL) [143], controlled parasitemia in vitro and in vivo of *P. falciparum* trophozoites at concentrations of 500, 250, and 125 μg/mL [144], and inhibited *P. berghei* ookinetes and gametocytes, in vitro, in *Anopheles stephensi* with a minimum inhibitory concentration of 25 μM [145]. Apamin reduces young *P. falciparum* trophozoites in vitro (MIC = 1 to 250 μg/mL). This peptide blocks potassium receptor channels and causes the parasite to become unviable [146].

#### 5.5.2. Phospholipase A from Bee Venoms

Phospholipase A was more related to antiplasmodial activity in bee venoms. PLA_5_ presented in vitro activity against *P. knowlesi*, with a significant reduction in intraerythrocytic growth at concentrations of 2–4 μM [147]. PLA_3_ presented in vitro activity against *P. falciparum* and reduced the level of parasitemia due to it preventing the development of the schizont [148]. Other PLA_3_s acted to inhibit parasite development, especially by reducing oocyst development and being active against *P. berghei* and *P. gallinaceum* [149,150] and inhibiting intraerythrocytic growth of *P. falciparum* [110,151,152]. PLA_4_ reduced the development of *P. falciparum* schizonts and trophozoites in in vitro tests [153]. PLA_2_ inhibited the development of young trophozoites in cells infected with *P. falciparum* (IC_50_ = 1.1 × 10^6^ μg/mL) [154].

**Table 4 toxins-15-00375-t004:** Toxins from venomous animals that have shown activity against *Plasmodium* spp.

Taxon	Family	Species	Chemical Class	Substance	Target Species	Development Stage	Magnitude of Activity		Model	Against	Ref.
							IC_50_ (μg/mL)	MIC (g/mL)			
Anurans	Bufonidae	*Rhinella marina*	Bufadienolide	Telocinobufagin	*P. f.*	Trophozoites	1.28	ND	In vitro	CQ-resistant strain	[95]
	Bufonidae	*Rhaebo guttatus*	Bufadienolide	Telocinobufagin	*P. f.*	Trophozoites	1.28	ND	In vitro	CQ-resistant strain	[95]
	Bufonidae	*Rhinella marina*	Alkaloid	Dehydrobufotenine	*P. f.*	Trophozoites	19.11	ND	In vitro	CQ-resistant strain	[122]
	Bufonidae	*Rhinella marina*	Bufadienolide	Marinobufotoxin	*P. f.*	Trophozoites	5.31	ND	In vitro	CQ-resistant strain	[122]
	Bufonidae	*Rhinella marina*	Bufadienolide	Marinobufagin	*P. f.*	Trophozoites	3.89	ND	In vitro	CQ-resistant strain	[122]
	Bufonidae	*Rhinella marina*	Bufadienolide	Bufalin	*P. f.*	Trophozoites	3.44	ND	In vitro	CQ-resistant strain	[122]
	Bufonidae	Not specified	Crude extract	Crude extract	*P. berghei*	ND	ND	ND	In vivo	Parasitemia	[98]
	Hylidae	*Phyllomedusa azurea*	Peptide	Phylloseptin-1	*P. f.*	Trophozoites	ND	128	In vivo	Parasite growth in erythrocytes	[128]
	Leptodactylidae	*Leptodactylus labyrinthicus*	Peptide	Oc-P1 (ocellatins)	*P. f.*	Trophozoites	26.71	ND	In vitro	CQ-resistant strain	[155]
	Pipidae	*Xenopus laevis*	Peptide	Magainin2	*An. Gambiae*	Zygotes, ookinetes, and merozoites	ND	0.5–1	In vitro	Parasite development in the mosquito	[156]
	Bufonidae	*Rhinella marina*	Crude extract	Crude extract	*P. f.*	Trophozoites	2.43	ND	In vitro	CQ-resistant strain	[120]
Spiders	Theraphosidae	*Psalmopoeus cambridgei*	Peptide	Psalmopeotoxin I (PcFK1)	*P. f.*	Intraerythrocytic cycle	116	ND	In vitro	Artemisinin-resistant strain	[157]
	Theraphosidae	*Psalmopoeus cambridgei*	Peptide	Psalmopeotoxin I (PcFK1)	*P. f.*	Trophozoites	1.59	ND	In vitro	Parasite growth in erythrocytes	[136]
	Theraphosidae	*Psalmopoeus cambridgei*	Peptide	Psalmopeotoxin II (PcFK2)	*P. f.*	Trophozoites	1.15	ND	In vitro	Parasite growth in erythrocytes	[136]
	Theraphosidae	*Psalmopoeus cambridgei*	Peptide	Psalmopeotoxin II’ (PcFK2’)	*P. f.*	Trophozoites	9.2	ND	In vitro	Parasite growth in erythrocytes	[158]
	Theraphosidae	*Acanthoscurria gomesiana*	Peptide	Gomesin	*P. f.*	Intraerythrocytic cycle	75.8–86.6	ND	In vitro	Parasite growth in erythrocytes	[159]
Scorpions	Scorpionidae	*Pandinus imperator*	Peptide	Scorpine	*P. berghei*	Ookinetes and gametes	ND	50 μM (fertilization); 3 μM (ookinete)	In vitro	Development of the para-site	[97]
	Buthidae	*Mesobuthus eupeus*	Peptide	Meucin-24	*P. f.*/*P. berghei*	Inhibi-ting the erythrocyte development	ND	40%;10 a 20 μM (inhibiting the development)	In vitro	Parasite growth in erythrocytes	[139]
	Buthidae	*Mesobuthus eupeus*	Peptide	Meucin-25	*P. f.*/*P. berghei*	Inhibiting the erythrocyte development	ND	50%;10 a 20 μM (inhibiting the development)	In vitro	Parasite growth in erythrocytes	[139]
	Vaejovidae	*Vaejovis mexicanus*	Peptide	VmCT1-NH_2_	*P. gallinaceum*	Sporozoites	0.49	ND	In vitro	Dead-cell staining	[140]
	Vaejovidae	*Vaejovis mexicanus*	Peptide	[Arg]3-VmCT1-NH_2_	*P. gallinaceum*	Sporozoites	0.57	ND	In vitro	Dead-cell staining	[140]
	Vaejovidae	*Vaejovis mexicanus*	Peptide	[Arg]7-VmCT1-NH_2_	*P. gallinaceum*	Sporozoites	0.51	ND	In vitro	Dead-cell staining	[140]
	Vaejovidae	*Vaejovis mexicanus*	Peptide	[Arg]11-VmCT1-NH_2_	*P. gallinaceum*	Sporozoites	>1.6	ND	In vitro	Dead-cell staining	[140]
Snakes	Viperidae	*Bothrops asper*	Crude extract	Crude extract	*P. f.*	Intra-erythrocytic cycle	0.13	ND	In vitro	CQ-resistant strain	[96]
	Viperidae	*Bothrops asper*	Enzyme	Fração V (Phospholipase A_2_)	*P. f.*	Intra-erythrocytic cycle	1.42	ND	In vitro	CQ-resistant strain	[96]
	Viperidae	*Bothrops asper*	Homologous	Fração VI (Homologo Phospholipase A_2_)	*P. f.*	Intra-erythrocytic cycle	323.35	ND	In vitro	CQ-resistant strain	[96]
	Elapidae	*Bungarus fasciatus*	Peptide	LZ1	*P. f.*	Intra-erythrocytic cycle	3.045	ND	In vitro	CQ-resistant strain	[107]
	Elapidae	*Bungarus fasciatus*	Peptide	LZ1	*P. berghei*	Intra-erythrocytic cycle	ND	39% (4 mg/kg), 35% (8 mg/kg) e 24% (12 mg/kg)	In vivo	CQ-resistant strain	[107]
	Viperidae	*Bothrops marajoensis*	Enzyme	BmajPLA_2_-II	*P. f.*	Intra-erythrocytic cycle	6.41	ND	In vitro	CQ-resistant strain	[112]
	Viperidae	*Agkistrodon halys*	Enzyme	Phospholipase A_2_ (IIA)	*P. f.*	Intraerythrocytic development	82.3	ND	In vitro	Parasite growth in erythrocytes	[110]
	Elapidae	*Naja mossambica mossambica*	Enzyme	Phospholipase A_2_ (IA)	*P. f.*	Intraerythrocytic development	0.023	ND	In vitro	Parasite growth in erythrocytes	[110]
	Elapidae	*Naja scutatus scutatus*	Enzyme	Phospholipase A_2_ (IA)	*P. f.*	Intraerythrocytic development	2.6	ND	In vitro	Parasite growth in erythrocytes	[110]
	Viperidae	*Vipera ammodytes*	Enzyme	Phospholipase A_2_ (IIA)	*P. f.*	intraerythrocytic development	2.8	ND	In vitro	Parasite growth in erythrocytes	[110]
	Elapidae	*Naja naja oxiana*	Extract	Fraction 4	*P. f.*	Intraerythrocytic development	0.368	ND	In vitro	Parasite growth in erythrocytes	[104]
	Elapidae	*Naja naja oxiana*	Extract	Fraction 4	*P. berghei*	Intraerythrocytic development	ND	70%(5 mg/kg); 50%(2.5 mg/kg); 30%(1 mg/kg)	In vivo	Parasite growth in erythrocytes	[105]
	Viperidae	*Bothrops brazili*	Metalloproteinase	BbMP-1	*P. f.*	Intra-erythrocytic development	3.2	ND	In vitro	Parasite growth in erythrocytes	[160]
	Viperidae	*Crotalus durissus*	Peptide	Crotamine	*P. f.*	Intra-erythrocytic development	1.87	ND	In vitro	CQ-resistant strain	[106]
	Viperidae	*Crotalus durissus cumanensis*	Enzyma (Fraction 1)	Crotoxin (Phospholipase A_2_)	*P. f.*	Intra-erythrocytic development	0.17	ND	In vitro	CQ-resistant strain	[111]
	Viperidae	*Crotalus durissus cumanensis*	Enzyme (Fraction 2)	Crotoxin (Phospholipase A_2_)	*P. f.*	Intra-erythrocytic development	0.76	ND	In vitro	CQ-resistant strain	[111]
	Viperidae	*Crotalus durissus cumanensis*	Enzyme	Crotoxin B (Phospolipase A_2_)	*P. f.*	Intra-erythrocytic development	0.6	ND	In vitro	CQ-resistant strain	[111]
	Viperidae	*Crotalus durissus cumanensis*	Crude extract	Crude extract	*P. f.*	Intra-erythrocytic development	0.17	ND	In vitro	CQ-resistant strain	[111]
	Viperidae	*Bothrops asper*	Enzyme	BaspAc-II	*P. f.*	Intra-erythrocytic development	2.46	ND	In vitro	CQ-resistant strain	[114]
	Viperidae	*Bothrops asper*	Enzyme	BaspAc-IV	*P. f.*	Intra-erythrocytic development	0.019	ND	In vitro	CQ-resistant strain	[114]
	Elapidae	*Micrurus spixii*	Crude extract	Crude extract	*P. f.*	Intra-erythrocytic development	≤0.78	ND	In vitro	CQ-resistant strain	[103]
	Viperidae	*Bothrops diporus*	Enzyme	BdTX-I (Phospholipase A_2_)	*P. f.*	Intra-erythrocytic development	2.44	ND	In vitro	CQ-resistant strain	[113]
	Viperidae	*Bothrops diporus*	Enzyme	BdTX-II (Phospholipase A_2_)	*P. f.*	Intra-erythrocytic development	0.0153	ND	In vitro	CQ-resistant strain	[113]
	Viperidae	*Bothrops diporus*	Enzyme	BdTX-III (Phospholipase A_2_) Homologo	*P. f.*	Intra-erythrocytic development	0.59	ND	In vitro	CQ-resistant strain	[113]
	Viperidae	*Crotalus adamanteus*	Enzyme	Phospholipase A_2_	*P. gallinaceum*	Oocyst formation	ND	ND	In vitro	ND	[109]
Bees	Not applicable	Not applicable	Enzyme	Phospholipase A_2_	*P. f.*	Young trophozoites	1.1 × 10^−6^	ND	In vitro	Intra-erythrocytic growth	[154]
	Not applicable	Not applicable	Enzyme	Phospholipase A_3_	*P. f.*	Tropho-zoites	1.69 × 10^−5^	ND	In vitro	Intraerythrocytic growth	[110]
	Not applicable	Not applicable	Enzyme	Phospholipase A_2_	*P. f.*	Mature trophozoites	ND	ND	In vitro	Intraerythrocytic growth	[155]
	Apidae	*Apis mellifera*	Enzyme	Phospholipase A_3_	*P. berghei*	Oocysts	ND	ND	In vitro	Development of the parasite	[149]
	Not applicable	Not described	Enzyme	Phospholipase A_3_	*P. gallinaceum*	Oocysts	ND	ND	In vitro	Development of the parasite	[150]
	Apidae	*Apis mellifera*	Peptide	Melittin	*P. f.*	Not specified	10	ND	In vitro	Intraerythrocytic growth	[143]
	Not applicable	Not applicable	Enzyme	Phospholipase A_3_	*P. f.*	Not specified	ND	ND	In vitro	Intraerythrocytic growth	[151]
	Not applicable	Not applicable	Enzyme	Phospholipase A_3_	*P. f.*	Schizonts	ND	ND	In vitro	Parasitemia	[148]
	Not applicable	Not applicable	Enzyme	Phospholipase A_4_	*P. f.*	Trophozoites and schizonts	ND	ND	In vitro	Parasitemia	[153]
	Not applicable	Not applicable	Peptide	Melittin	*P. f.*	Trophozoites and schizonts	ND	ND	In vitro	Development of the parasite	[94]
	Not applicable	Not applicable	Peptide	Melittin	*P. f.*	Trophozoites	ND	ND	In vitro and in vivo	Parasitemia	[144]
	Not applicable	Not applicable	Enzyme	Phospholipase A_5_	*P. knowlesi*	Trophozoites	ND	ND	In vitro	Intraerythrocytic growth	[147]

Abbreviations: IC_50_—Half-maximal inhibitory concentration; MIC—Minimum inhibitory concentration; Not determined (ND); NMR 1H—Hydrogen-1 nuclear magnetic resonance; NMR 13C—Carbon-13 nuclear magnetic resonance; F-moc—Fluorenylmethoxycarbonyl protecting group; HPLC—High-performance liquid chromatography; MALDI-TOF—Matrix-assisted laser desorption/ionization-time of flight; TOF MS—Time-of-flight mass spectrometry; RP-HPLC—Reversed-phase HPLC; SDS-PAGE—Sodium dodecyl sulfate-polyacrylamide gel electrophoresis; Ref.—References; *P. f*.—*Plasmodium falciparum*; *P. v*.—*Plasmodium vivax*; CQ—chloroquine.

## 6. Conclusions and Future Perspectives

Drug therapies and vector control via insecticides are respectively the most used methods for the treatment and control of malaria; however, several studies have shown resistance of some *Plasmodium* species to the drugs that are recommended for their treatment. In view of this, it is necessary to carry out studies to discover new antimalarial molecules as lead compounds for the development of medicines. As such, in the last few decades, animal venoms have attracted attention for being a potential source for new antimalarial molecules. 

In this review, we evidenced 50 substances, 4 fractions and 7 toxins extracted from the venoms of animals. Anurans, snakes, spiders, scorpions and bees have been studied, and all of them have shown immeasurable antimalarial activities against *Plasmodium* spp., acting in distinct phases of its biological cycle and with its consequent inhibition. However, it is emphasized that more studies should be carried out in order to unravel the mechanism of action of the toxins in the inhibition of *Plasmodium* spp., as they represent a major milestone in the face of the resistance to current antimalarial drugs.

## Figures and Tables

**Figure 1 toxins-15-00375-f001:**
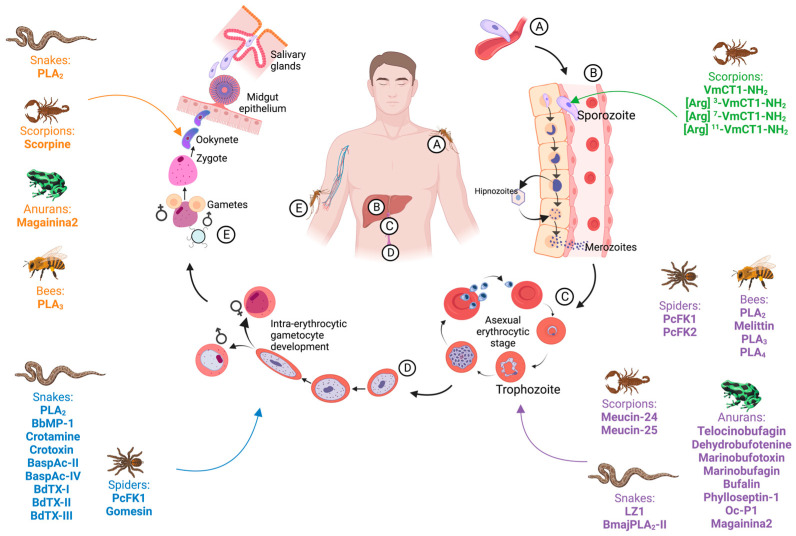
*Plasmodium* life cycle, showing the points at which toxins from animal venoms can act. Inoculation of sporozoites into the host’s epidermis (**A**); hepatic and pre-erythrocytic stage (**B**); erythrocytic stage (**C**); intraerythrocytic phase and gametocyte development (**D**); *Anopheles* spp. infection and parasite development in the mosquito midgut (**E**); male gamete (♂); female gamete (♀).

**Figure 2 toxins-15-00375-f002:**
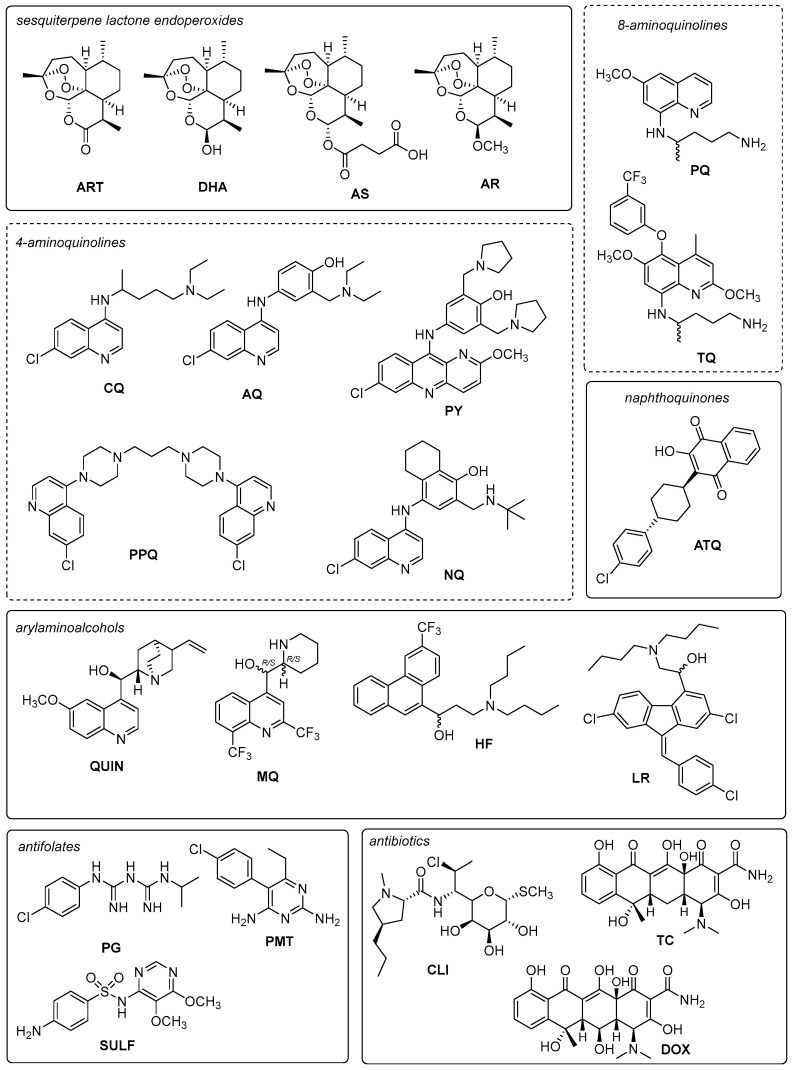
Currently used antimalarial drugs.

**Figure 3 toxins-15-00375-f003:**
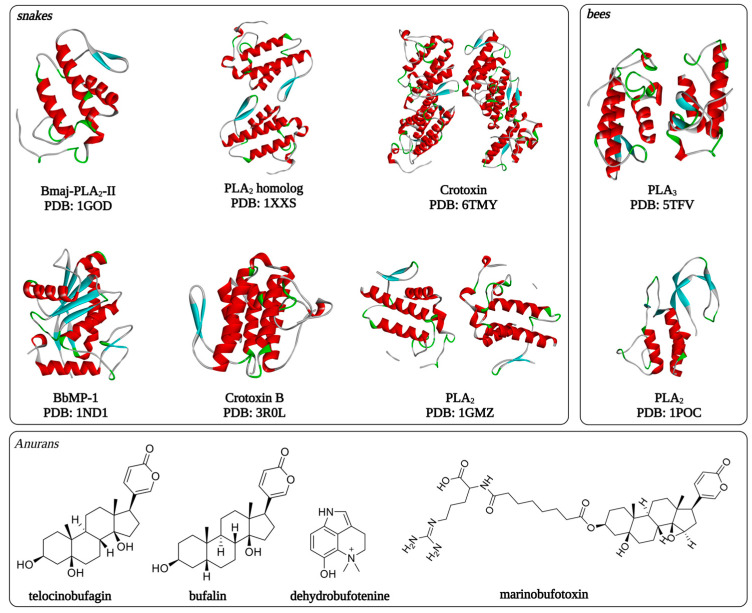
Three-dimensional structures of animal venom toxins with antiplasmodial activity.

## Data Availability

No new data were created or analyzed in this study. Data sharing is not applicable to this article.

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
