# Peer review of "Toxins from Animal Venoms as a Potential Source of Antimalarials: A Comprehensive Review"

_toxins, 2023, doi:10.3390/toxins15060375_

Round 1

Reviewer 1 Report

This review describes toxins from venomous animals with anti-plasmodium properties as potential sources of new anti-malarial molecules and provides good ideas for current anti-malarial drug resistance. However, there are some problems with the review:

1.The title of the review is "Toxins from venomous animals against Plasmodium spp", but the real introduction begins in the second half, and the previous introduction is too long;

2.Table 25 mentioned in the article has not been found;

3.Plasmodium berghei and Plasmodium gallinaceum are mentioned in the fifth part of the article, which were not mentioned in the previous description;

4.The illustration in Figure 2 is highly repetitive with the description in the above part, so it is suggested to simplify;

5.A small number of references are too old.

Author Response

Dear reviewer, thank you for appreciating the manuscript. Attached is the reply letter. yours sincerely

Reviewer 2 Report

The manuscript “Toxins from venomous animals against Plasmodium spp. summarizes the current knowledge regarding malaria, the protozoan Plasmodium causing the disease, the vectors transmitting the disease, treatment options, resistance mechanisms, and finally toxins with possible anti-Plasmodium activity.

 Overall, the article offers a comprehensive, often exhausting overview regarding all the aforementioned topics, is well structured, and is of great interest to the general public. However, there are several issues that should be resolved for the manuscript to be acceptable for publication.

1)      The title suggests that the topic of the manuscript are the toxins from venoms active against Plasmodium spp. However, the manuscript in its current form includes many other details, not directly related to the stated topic. Mainly, half of the manuscript is in fact not about toxins and venoms, but about Plasmodium, malaria, and its treatment. While this information is definitely of great interest, and in some degree required to better understand the mechanism of action of toxins, it is overwhelming in its current form. I propose two alternative solutions for this issue. a) Change the title of the manuscript to reflect the fact that it is not only about toxins, but a broader overview of malaria and its treatment; or b) truncate the first part of the manuscript to a far more succinct overview of the presented topics (e.g.: shorten the subsections regarding Plasmodium to 2-3 paragraphs, remove “Vectors” column from table 1, possibly all of Table 1 and 2, as they are unrelated to toxins, shorten or remove treatment and resistance sections, etc.)

2)      Extensive editing of English language required. In its current form, the manuscript is not suitable for publication. For details see “Comments on the Quality of English Language”.

3)      Some personal input and interpretation should be given. The relevant toxins are presented, with results taken from the respective reference, but there is no overview offered. Are there any outstanding toxins for the treatment of malaria among those presented? If yes, what differentiates them from the rest? Are there any clinical trials planned / underway with compounds based on these toxin structures? If not, what are the limitations? Based on the presented results, there are several promising lead molecules, are they considered for development? A more in-depth discussion about potential mechanisms of action would add value to the manuscript, rather than presenting raw MIC data.

4)      References contain an unusually large number of online sources. If possible, try to reference source documents, rather than websites presenting the content of those documents. Furthermore, in some places excessive referencing is used, with 4-5 sources mentioned for an information. References should be limited to those relevant, there is no need to inflate the number of sources cited. Lastly, the structure of the manuscript reflects in the references section as well – only approximately half of the sources are related to toxins – meaning 70-80 cited sources are unrelated to the stated title (see the solutions proposed previously).

Some minor errors that should be corrected:

1)      Line 63: Anopheles and Plasmodium should be uppercase.

2)      Line 165: “Table 25” text should be deleted.

3)      Section 5.5.1.: Crude snake venoms. There are two peptides presented in this section, which should not be considered “crude venom”.

4)      Amino acid sequences of peptides do not add additional information. They should be deleted from the text body.

The manuscript contains many grammar or syntax errors. An improvement of the English language is necessary. There are too many errors to mention them all, but a few locations where the text needs improving / rephrasing: lines 6-7, 12-13, 44-47, 57-60, 130-132, 239-241, 245-249, 254-258 etc.

Author Response

Dear reviewer, thank you for appreciating the manuscript. Attached is the reply letter. Best regards

Round 2

Reviewer 2 Report

The scope of a peer review is not only to correct errors, but in the same time to improve the quality of an article. The initial recommendations were given with the well-meant intention to increase the quality of the manuscript. Except for English language editing, all relevant recommendations have been rejected by the authors.

Author's Response:  The inclusion of other topics around plasmodium are extremely important to lead any reader to understand the great problem we face today, from distribution of Plasmodium and its respective vectors by region, treatment, resistance to treatment illustrating the different mechanisms of resistance in order to illustrate that in fact there is a need to search for new therapeutic alternatives. as for the theme of the manuscript, we decided to keep it the same.

Response: Indeed, the presented topics are of great interest. However, the title of the manuscript remains misleading, as approximately half of the manuscript is not in fact about toxins. I maintain my recommendation to adapt the title with the content of the manuscript.

Author's Response: We review and correct the inconveniences. thanks for the observation.

Response: The English editing has been performed as requested.

Author's Response: We would like to be able to succinctly satisfy these questions and suggestions, however, what was presented in this review regarding toxins against Plasmodium is what exists. thus, we believe from this integrative review that more curiosities may arise around toxins against Plasmodium, triggering further therapeutic studies.

Response: The initial comment included several recommendations to be considered. None were included in the manuscript.

Author's Response: The high number of references corresponds to citations throughout the text. all references here are the original sources from which the information was taken.

Response: Indeed, all references correspond to citations throughout the text. The initial recommendation mentioned a large number of sources for the same information, and the overuse of online sources. Neither of these issues was addressed in the author's response

Author's Response: Please see again, the name of the section is Peptides from bee venoms (line 433)

Response: Indeed, there was a typo in the section number. It was referencing Section 5.1.1.: Crude snake venoms. A brief reading of the mentioned section based on its title would have revealed where the comment was pointing at.

Author Response

Good afternoon, dear reviewer, we would like to express our gratitude for the rich suggestions. Attached is the reply letter. Best regards
